# Triangulation Residual Loss for Data-efficient 3D Pose Estimation

**Jiachen Zhao**
Tsinghua University
Beijing, China 100084
zhao_jiachen@163.com

**Tao Yu**[†]
Tsinghua University
Beijing, China 100084
ytrock@126.com

**Liang An**
Tsinghua University
Beijing, China 100084
al17@mails.tsinghua.edu.cn

**Yipeng Huang**
Beijing Institute of Technology
Beijing, China 100081
huangyipeng@bit.edu.cn

**Fang Deng**
Beijing Institute of Technology
Beijing, China 100081
dengfang@bit.edu.com

**Qionghai Dai**[†]
Tsinghua University
Beijing, China 100084
qhdai@mails.tsinghua.edu.cn

## Abstract

This paper presents Triangulation Residual loss (TR loss) for multiview 3D pose estimation in a data-efficient manner. Existing 3D supervised models usually require large-scale 3D annotated datasets, but the amount of existing data is still insufficient to train supervised models to achieve ideal performance, especially for animal pose estimation. To employ unlabeled multiview data for training, previous epipolar-based consistency provides a self-supervised loss that considers only the local consistency in pairwise views, resulting in limited performance and heavy calculations. In contrast, TR loss enables self-supervision with global multiview geometric consistency. Starting from initial 2D keypoint estimates, the TR loss can fine-tune the corresponding 2D detector without 3D supervision by simply minimizing the smallest singular value of the triangulation matrix in an end-to-end fashion. Our method achieves the state-of-the-art 25.8mm MPJPE and competitive 28.7mm MPJPE with only 5% 2D labeled training data on the Human3.6M dataset. Experiments on animals such as mice demonstrate our TR loss's data-efficient training ability.

## 1 Introduction

Multiview 3D human/animal pose estimation is a crucial and challenging task with wide application in mixed reality telepresence, animation, and neuroscience. Most existing animal pose estimation (APE) methods are transferred from the highly developed human pose estimation (HPE) methods [1, 2]. The success of HPE depends on large-scale datasets [3–5] and high-quality 3D modelS [6]. However, these conditions are not met in APE tasks. On the one hand, labeling large-scale datasets for each species requires significant user effort but cannot obtain the same commercial benefits as HPE. On the other hand, obtaining 3D ground truth via sticky marks is not feasible since the animal's body could be tiny and animals tend to destroy the markers.

Triangulation is a crucial approach for multiview pose estimation due to its efficiency and effectiveness. However, traditional triangulation involves a non-differentiable argmax and RANSAC process

disabling the end-to-end training. The learnable triangulation methods [7–10] have been proposed to perform triangulation in an end-to-end fashion. Its typical pipeline is (1) generating the 2D joint heatmaps for each view, (2) using soft-argmax to get 2D joint coordinates from the heatmaps, and (3) performing triangulation to estimate 3D joint coordinates. Finally, the MAE loss between the ground truth and the predicted 3D positions is used to train the model as shown in Figure 1 (a). Several works introduce learnable weights [9], and stochastic frameworks [10] to improve performance and generalization. However, 3D supervision needs large-scale labeled data which is usually unavailable for APE.

Some papers [11–14] introduce epipolar geometry as a supervisory signal to eliminate 3D ground truth. Its key idea is shown in Figure 1 (b). The epipolar geometry restricts that the 2D keypoint in view 2 must lie in the corresponding epipolar line $l'_1$ from view 1. To enable the epipolar constraint to be trained end-to-end, Yao et al. [12] propose the epipolar divergence which measures the distribution distance between epipolar lines. These epipolar-based methods pursue local consistency in pairwise views, so most of them involve trivial pairing and reprojection processes. They may also suffer from heavy computation when the view number is large.

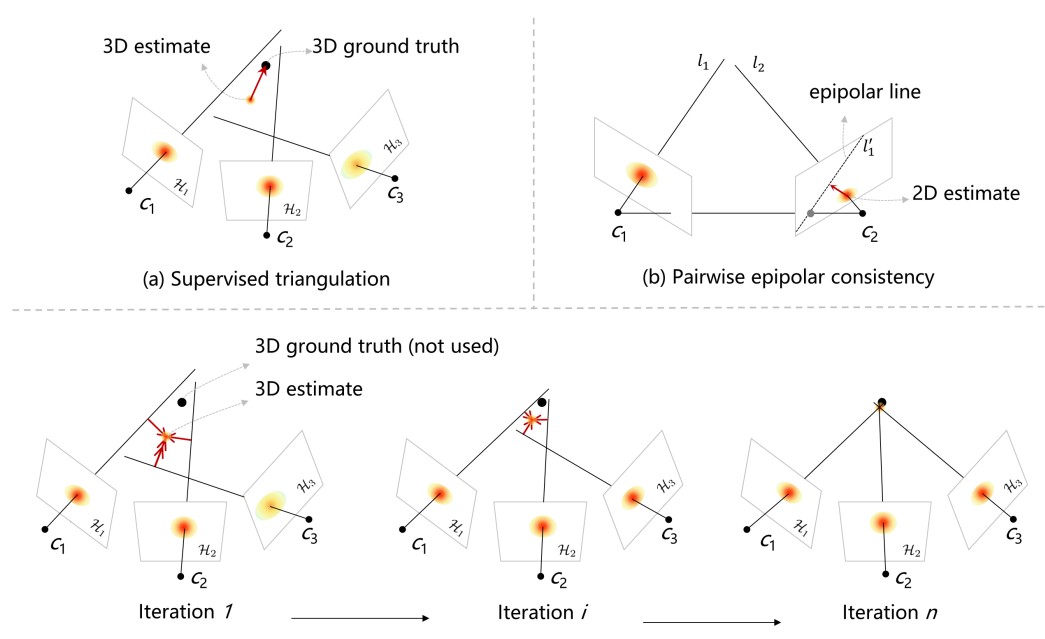

(a) Supervised triangulation    (b) Pairwise epipolar consistency

(c) Global geometric consistency via TR loss

Figure 1: (a) Supervised triangulation trained with MSE loss between 3D estimation and 3D ground truth. (b) Unsupervised triangulation with pairwise epipolar consistency. (c) Self-supervised triangulation with TR loss. The TR loss realigns the predicted heatmap locations to the accurate point using multi-view consistency. Geometrically, TR loss can be viewed as the sum of distances between 3D estimate and view rays as shown in the figure. Algebraically, TR loss can be simply formulated as the smallest singular value of the triangulation matrix.

The paper tackles these challenges by leveraging global multiview geometric consistency. The core of our methods is a simple but powerful loss function, called Triangulation Residual Loss (TR loss), which is defined as the sum of distances between the 3D estimate and view rays as shown in Figure 1 (c). Supervised methods aim to minimize the absolute error between the 3D estimate and 3D ground truth. Differently, our goal is to minimize the residual between the 3D estimate and 2D observations, where 2D observation is represented by the view ray from the camera center to the 2D keypoint in the image. In every single iteration in Figure 1 (c), we estimate the optimal 3D estimate by triangulation, but there must be residuals, i.e. the distances between 3D estimate and view rays, since the 3D estimate doesn't lie in all view rays. Iterative minimizing the residual will lead the 3D estimate to converge to a geometrically consistent position. Moreover, we use algebraic errors to approximate geometric errors, TR loss is simplified to the smallest singular value of the triangulation matrix without any reprojection processes.

The contributions can be summarized as follows: (1) A simple yet effective loss function, TR loss, for multiview 3D pose estimation. It achieves data-efficient training by using the multiview geometrical consistency without 3D supervision and any reprojection processes. It can be simply formulated as minimizing the smallest singular value of the triangulation matrix. (2) A new benchmark for laboratory mouse 3D pose estimation based on existing datasets (CalMS21 and Dannce) and THmouse data collected by ourselves. (3) Our method achieves state-of-the-art MPJPE performance for single-frame 3D pose estimation on the Human3.6M dataset and obtains competitive performance with only 5% 2D labeled training data. Code has been made available at https://github.com/zhaojiachen1994/Triangulation-Residual-Loss.

## 2 Related works

**Learnable triangulation** Triangulation is popular in 3D multiview animal pose estimation [15, 16]. To overcome the non-differentiability of the vanilla triangulation (RANSAC), the soft and differentiable versions of RANSAC have been proposed [7, 8, 10]. The core of these methods is to select high-confidence 2D estimates via the probabilistic learning scheme. Karim et al. [9] utilize the learnable weights to softly select the 2D estimates. Kristijan et al. [10] fix the pretrained 2D detector and use the subsequent hypothesis selection process to finetune the 3D pose. Therefore, this method heavily relies on the performance of the pretrained 2D detector. Some volumetric approaches [17, 18] also achieve great performance on 3D pose estimation. However, all the above methods require ground-truth 3D labels during training, so they are not suitable for APE. Some methods leverage the epipolar geometry to learn feature consistency [11, 12] or geometry consistency [13, 14], so that the 3D supervision is no longer required. However, all these methods involve complex reprojection and alignment in pairwise views. Therefore, they only consider the local pairwise consistency and become inapplicable when the number of cameras is large.

**Animal pose estimation** The physical behavior of animals is the most intuitive external manifestation of neural activity. However, observing animal behavior is one of the most time-consuming operations in biology or neuroscience research. Experimenters need to spend several hours observing and recording the behavior of mice only to complete a trial. A research topic often requires hundreds of trials. Fortunately, vision-based keypoint detection methods provide an efficient tool to measure the behaviors of lab animals. However, labeling data under different experimental paradigms is still labor-intensive. Therefore, domain-adaptation [19–22] methods have been introduced to APE. Li et al. [19] and Cao et al. [22] design pseudo-label generation approaches to expand labeled data during training. Jiang et al. [20] and Mu et al. [21] utilize synthetic animal images as the source domain and reduce the domain shift to predict animal pose in real images. RegDA [23] extends the disparity discrepancy theory from classification to keypoint detection and improves the cross-domain keypoint detection performance. The above methods are limited to 2D pose estimation, which loses key information in the analysis of experimental animal behavior. Further, this paper focuses on multiview 3D APE with only a few 2D labels.

Different from existing learnable triangulation methods, our method achieves global 3D geometry consistency without 3D supervision and reprojection operations. Moreover, the loss can be simply formulated as minimizing the smallest eigenvalue of the triangulation matrix.

## 3 Methods

### 3.1 Problem formulation

We assume a few 2D labeled pose data are given and aim to estimate the 3D pose from multiview images. The 2D labeled data $\mathcal{D}_l$ can be an existing large-scale dataset or a few annotations under the same dataset, which contains input images $\boldsymbol{I}_i \in \mathbb{R}^{H \times W \times 3}$ and the corresponding ground-truth 2D keypoint coordinates $\boldsymbol{J}_i^{2d} \in \mathbb{R}^{J \times 2}$ for all keypoints, where $J$ is the number of keypoints. We can train a 2D keypoint detector on the labeled data $\mathbb{R}^{H \times W \times 3} \mapsto \mathbb{R}^{J \times 2}$. The target multiview images $\mathcal{D}_u$ are simultaneously captured from $C$ cameras but unlabeled, where $\boldsymbol{I}_{c,i}$ is the image captured by the $c$th camera at the $i$th scene. Our goal is to train a 3D keypoint detector $\mathbb{R}^{C \times H \times W \times 3} \mapsto \mathbb{R}^{J \times 3}$ to predict the global 3D positions for $\mathcal{D}_u$. Throughout this paper, we use $\boldsymbol{x}_{c,j}$ to indicate the 2D coordinate of $j$th joint in $c$th view and use $\boldsymbol{X}_j$ to indicate the 3D coordinate of $j$th joint.

## 3.2 Model structure

Figure 2 presents the framework of our model including a backbone network $\phi$, a heatmap head $\boldsymbol{h}$, a confidence head $\boldsymbol{f}$, a differentiable triangulation head and an optional domain discriminator $\psi$.

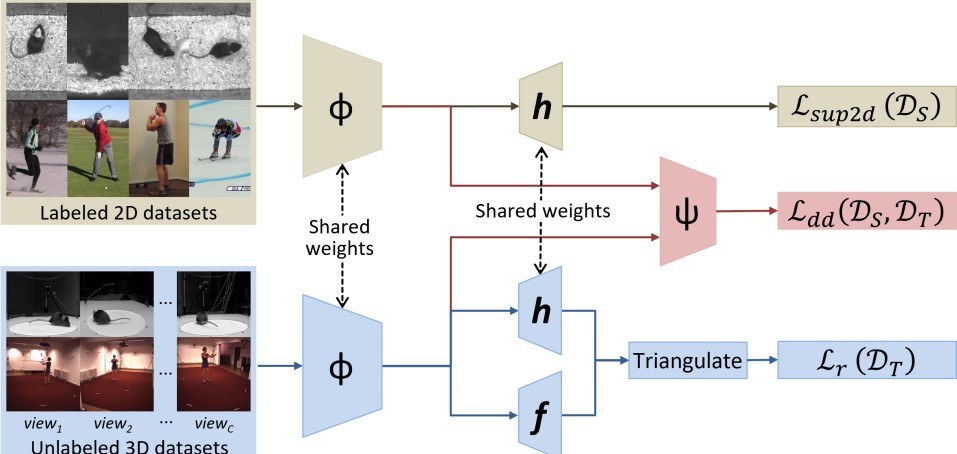

Figure 2: The framework of our method containing a backbone network $\phi$, a heatmap head $\boldsymbol{h}$, a confidence head $\boldsymbol{f}$, a differentiable triangulation head and an optional domain discriminator $\psi$.

The input data contains several 2D labeled images and unlabeled target multiview images. All input images are fed into a CNN-based backbone $\phi$ to generate feature maps. Following the backbone, a heatmap head $\boldsymbol{h}$ transforms the intermediate feature maps to keypoint heatmaps $\mathbb{R}^{H \times W \times 3} \mapsto \mathbb{R}^{H' \times W' \times J \times 2}$. Then the keypoint heatmaps are transformed into 2D coordinates via a differentiable process including spatial softmax in Eq. (1) and soft-argmax Eq. (2):

$$\text{Softmax}(\mathcal{H}_{c,j}) = \exp(\mathcal{H}_{c,j})/(\sum_{h'}^{H'} \sum_{w'}^{W'} \exp(\mathcal{H}_{c,j}(h', w'))) \tag{1}$$

$$\hat{\boldsymbol{x}}_{c,j} = \sum_{h'}^{H'} \sum_{w'}^{W'} [h', w'] \odot \mathcal{H}_{c,j}(h', w') \tag{2}$$

Here, $\mathcal{H}_{c,j}$ is the heatmap for $j$th joint in $c$th view. $H'$ and $W'$ are the heatmap height and width. $h'$ and $w'$ are the spatial indexes across heatmap height and width. $\odot$ means the element-wise product. We further denote the combination of Eq. (1) and Eq. (2) as $\boldsymbol{\kappa}$, so the 2D keypoint detector can be formulated as

$$\hat{\boldsymbol{x}}_{c,j} = (\boldsymbol{\kappa} \circ \boldsymbol{h} \circ \phi(\boldsymbol{I}_c))_j \tag{3}$$

Parallel with the heatmap head, a confidence head $\boldsymbol{f}$ produces the triangulation confidence for each keypoint in each view $\mathbb{R}^{H \times W \times 3} \mapsto \mathbb{R}^J$:

$$w_{c,j} = (\boldsymbol{f} \circ \phi(\boldsymbol{I}_c))_j \tag{4}$$

Following $\boldsymbol{h}$ and $\boldsymbol{f}$, a triangulation head estimates the 3D position $\boldsymbol{X}_j^*$ based on the multiview 2D estimates by minimizing the weighted sum of reprojection errors in all views:

$$\boldsymbol{X}_j^* = \arg \min_{\boldsymbol{X}_j} \sum_{c=1}^{C} w_{c,j} \cdot ||\hat{\boldsymbol{x}}_{c,j} - \boldsymbol{P}_c \cdot \boldsymbol{X}_j||^2 \tag{5}$$

where $|| \cdot ||$ is the L2 norm, $\boldsymbol{P}_c$ is the projection matrix of $c$th view and $w_{c,j}$ is the learnable weight to adjust the contribution of each view in the triangulation process so that the erroneous 2D estimates will have little effect. However, direct solving Eq.(5) is not differentiable, so we use the linear algebraic method, which is to solve the overdetermined system of equations on homogeneous $\boldsymbol{X}_j$:

$$\boldsymbol{w}_j \odot \boldsymbol{A}_j \cdot \boldsymbol{X}_j = 0 \tag{6}$$

where $A_j$ is the matrix composed of $\boldsymbol{P}_c$ and $\hat{\boldsymbol{x}}_{c,j}$. We further annotate $\widetilde{\boldsymbol{A}}_j = \boldsymbol{w}_j \odot \boldsymbol{A}_j$ as the triangulation matrix in this paper. The details about how to formulate $\widetilde{\boldsymbol{A}}_j$ can be found in Appendix A.1. The overdetermined equations Eq. (6) can be solved via SVD as $\widetilde{\boldsymbol{A}}_j = U\Sigma V^T$. The optimal estimate $\boldsymbol{X}_j^*$ is the last column of $V$. So far, supervised triangulation methods can train the model with MSE loss between the 3D estimate and 3D ground truth: $\mathcal{L}_{\text{sup}} = \left\| \boldsymbol{X}_j^* - \boldsymbol{X}_j^{\text{gt}} \right\|^2$. Differently, we eliminate the 3D supervision by using the TR loss presented in Section 3.3.

### 3.3 Triangulation residual loss

Figure 1 (c) presents the intuition of TR loss from a geometric perspective. In the first iteration, we can estimate a preliminary 3D estimate $\boldsymbol{X}^*$ by solving Eq. (6). Since the 2D detector is pretrained on limited data, there must be estimation errors in the 2D position and there may be some large deviations, such as $\mathcal{H}_3$. As a result, $\boldsymbol{X}^*$ is inaccurate and does not fall on view rays. In this paper, we minimize the sum of distances from $\boldsymbol{X}^*$ to each view ray so that $\boldsymbol{X}^*$ will converge to a reliable position. During iterative minimization, the learnable weight $\boldsymbol{w}_{c,j}$ is also updated in such a way that 2D estimates with less consistency (such as $\mathcal{H}_3$) are given smaller weights, and 2D estimates with more consistency (such as $\mathcal{H}_1$ and $\mathcal{H}_2$) are given larger weights. To sum up, the TR loss realigns the predicted heatmap locations to the accurate point using multi-view consistency.

From an algebraic perspective, the overdetermined Eq. (6) can not be fully established. There must be residual as $\left\| \widetilde{\boldsymbol{A}}_j \cdot \boldsymbol{X}_j^* - 0 \right\|$. Minimizing the sum of distances from $\boldsymbol{X}^*$ to each view ray can be reduced to minimize the

$$\mathcal{L}_{\text{res},j} = \left\| \widetilde{\boldsymbol{A}}_j \cdot \boldsymbol{X}_j^* \right\|^2 \tag{7}$$

Since $\boldsymbol{X}_j^*$ is the right singular vector corresponding to the smallest eigenvalue of $\widetilde{\boldsymbol{A}}_j$. If we omit the subscript $j$, $\mathcal{L}_{\text{res}}$ can be reformulated as

$$||\widetilde{\boldsymbol{A}}\boldsymbol{X}^*||^2 = ||\widetilde{\boldsymbol{A}}v_4||^2 = ||\Sigma_{i=1}^4 \sigma_i u_i v_i^T v_4||^2 = ||\sigma_4 u_4||^2 = ||\sigma_4||^2 \tag{8}$$

In Eq. (8), $\widetilde{A}$ with rank $= 4$ can be decomposed by SVD as $\widetilde{\boldsymbol{A}} = \Sigma_{i=1}^4 \sigma_i u_i v_i^T$, where $\sigma_1 \geq \sigma_2 \geq \sigma_3 \geq \sigma_4$ are the unit singular values of $\boldsymbol{X}^*$, $u_i$s and $v_i$s are the left and right singular vectors of $\widetilde{\boldsymbol{A}}$. According to the property of SVD, both $u_i$s and $v_i$s are unit vectors and orthogonal to each other, so Eq. (8) is established. Here, we directly use PyTorch's built-in auto-differential SVD function, while robust differentiable SVD [24] is also an alternative. Finally, $\mathcal{L}_{\text{res}}$ can be simply formulated as minimizing $\widetilde{\boldsymbol{A}}_j$'s smallest eigenvalue $\sigma_4$:

$$\mathcal{L}_{\text{res}} = \left\| \sigma_4 \right\|^2 \tag{9}$$

**Eliminate trivial solutions** A straightforward way to obtain the learnable weight in $\widetilde{\boldsymbol{A}}_j$ is through the sigmoid function. If so, the TR loss tends to assign high confidence (nearly 1) to only two views and assign low confidence (nearly 0) to other views, since the fewer views involved in triangulation, the smaller the error. The TR loss can no longer improve 2D estimates in low-confidence views. Moreover, all weights equal to 0 is a trivial solution for Eq. (7). So we set a threshold range for the confidence score to shrink it. More details about the threshold are discussed in Section 4.4.

### 3.4 Total loss functions

Since the TR loss only pursues geometric consistency, this may cause the 2D estimates to be non-semantic points instead of the correct joints. Therefore, we still need 2D supervision loss $\mathcal{L}_{\text{sup2d}}$ on the labeled 2D data $\mathcal{D}_l$. Same as in [25, 26], we utilize the pixel-wise MSE loss between the predicted heatmap $\hat{\mathcal{H}}_{i,j}$ and the ground-truth heatmap $\mathcal{H}_{i,j}$:

$$\mathcal{L}_{\text{sup2d}} = \sum_i^M \sum_j^J ||\mathcal{H}_{i,j} - \hat{\mathcal{H}}_{i,j}||^2 \tag{10}$$

If the labeled images $\mathcal{D}_l$ and the target multiview images $\mathcal{D}_u$ are from different domains, we can also add a domain discriminator $\psi$ trained with domain discrimination loss $\mathcal{L}_{dd}(\mathcal{D}_l, \mathcal{D}_u)$ to reduce domain discrepancy. Same as in [22, 27, 28], the domain discrimination loss is defined as cross-entropy loss and set to be adversarial to $\mathcal{L}_{sup2d}$ and $\mathcal{L}_{res}$:

$$\mathcal{L}_{dd} = -\sum_{i=1}^{N}(z_i \log(\hat{z}_i) + (1 - z_i)\log(1 - \hat{z}_i)) \tag{11}$$

where $z_i$ and $\hat{z}_i$ are the ground-truth and predicted domain labels, respectively. The total loss function is

$$\mathcal{L} = \mathcal{L}_{res} + \alpha\mathcal{L}_{sup2d} + \beta\mathcal{L}_{dd} \tag{12}$$

where $\alpha$ and $\beta$ are tradeoff hyperparameters. The ablation studies in Section 4.4 show that our proposed TR loss is the most critical component but the DD loss is dispensable.

## 4 Experiments

To evaluate the effectiveness of our proposed method, we conduct experiments on lab mouse pose estimation and human pose estimation. It should be noted that we only use 2D labels (not 3D) to train our model.

### 4.1 Implement details

We implement our model based on the mmpose library [29]. We apply the HRnet as the backbone, build the heatmap head with a convolution layer with a 3x3 kernel size, and build the confidence head with two convolution layers followed by three linear layers [512, 256, $J$] with a sigmoid activation function. The domain discriminator is an average pooling layer followed by three linear layers [512, 256, $J$] with a sigmoid activation function. The models are trained on 1 NVIDIA 3090 GPU with the Adam optimizer and an initial learning rate of $1e^{-5}$. The loss tradeoff parameters are set to 1, so all losses have the same effect.

### 4.2 Human Pose Estimation

**Datasets** We conduct human pose estimation experiments on the Human3.6M dataset. The Human3.6M dataset [3] contains 3.6 million frames of single-person activity captured by 4 synchronized cameras. Its ground-truth 3D joint position is obtained by a marker-based motion capture system with 10 IR sensors. We assume that only 2D labels in the training dataset are available to train a 2D detector. Then, we train the total model with TR loss on the 1% multiview unlabeled data and heatmap loss on the labeled training images. Therefore, our method does not require any additional annotations.

**Results** Table 1 presents the MPJPE metrics of our method and the comparison methods. All methods in Table 1 are based on single-frame, multiview images of the Human3.6M dataset, excluding some methods using additional labeled training data [9, 31] and single-view methods [32, 33]. Table 1 shows that our method achieves 1.1mm MPJPE than the existing best method. Moreover, our method achieves 28.7 MPJPE with only 5% training data to train the 2D detector. The MPJPE of naive triangulation is 69.9mm, indicating that the erroneous 2D estimates at certain views heavily af-

Table 1: MPJPE results on the Human3.6M dataset following Protocol 1

| Method | Supervision | Average |
|---|---|---|
| Baseline | 2D | 69.9 |
| Cross-view fusion [13] | 3D | 31.2 |
| Remelli et al. [17] | 3D | 30.2 |
| Epipolar transformers [14] | 2D | 26.9 |
| Kristijan et al. [10] | 3D | 29.1 |
| MTFT-Transformer [30] | 3D | 27.5 |
| TR loss (100%) | 2D | **25.8** |
| TR loss (5%) | 2D | 28.7 |

fect the final 3D estimates. Our method achieves significant improvements and even performs better than 3D supervised methods [13, 17, 10, 30]. These results demonstrate that the TR loss is capable of

reducing the negative effect of erroneous views and guiding the view rays converging to the ground truth (as the schematic in Figure 1 (c) shows).

## 4.3 Laboratory mouse pose estimation

**Datasets** Laboratory mouse pose estimation is a key task in neuroscience. However, there is still no high-quality large-scale multiview dataset, demonstrating the need for a data-efficient approach. We conduct experiments on three datasets to evaluate our method: CalMS21, Dannce, and THmouse (THM, a dataset we collected). CalMS21 [34] is a large-scale 2D mouse dataset containing 30,000 frames of two interacting mice captured by two cameras from top-view and front-view. Nine joints (top-view) and thirteen joints (front) of each mouse are annotated in each frame. Dannce dataset [35] consists of 1032 labeled frames recorded by six cameras around a single mouse. Our THmouse dataset has 1200 frames from 6 synchronized cameras (1 top-view and 5 around-views). We annotate the 2D ground truth and triangulate the 3D ground truth based on 2D labels. We conduct six pose estimation tasks: CalMS21-Dannce, CalMS21-THmouse, Dannce-THmouse, THmouse-Dannce, Dannce, and THmouse. In intra-dataset experiments, we split the training/test set by 60%/40% and train models on the training set. In cross-dataset experiments, we also split the target dataset into training data and test data by 60%/40%. Then we use the 2D ground truth on the source dataset to compute heatmap loss and use the target training data to compute the TR loss. We select the shared joint across datasets to evaluate methods and use MPJPE as the evaluation metric. Details about the joint definition can be found in Appendix A.4.

**Results** Table 2 shows the MPJPE results on mouse pose estimation. The baseline directly triangulates the 3D estimates via SVD based on the 2D detection results of pretrained HRnet. RANSAC is implemented based on the Anipose toolkit [36]. Both methods above are based on the results of a pretrained 2D detector without other fine-tuning, so the performance is relatively poor. DeeplabCut [15] is the most popular animal pose estimation method. GeneralTriang [10] is a stochastic framework for generalizable pose estimation, but it relies on the predicted 2D results, so it can not update the 2D detector in an end-to-end fashion. The Sup3d* model [9] is finetuned with MSE loss between the 3D estimate and 3D ground truth. TR model even achieves better results on some tasks than Sup3d* even without using the 3D ground truth. TR+DD model adds the domain discrimination loss to reduce domain discrepancy in cross-dataset tasks, slightly improving performance. Training the model with both TR loss and 3D supervision loss (TR+Sup3d*) can lead every view ray to focus on the 3D ground truth, so it achieves the best MPJPE results for all tasks.

Table 2: MPJPE results on mouse pose estimation. * means the method used 3D ground truth during training. The unit is mm.

|  | Dannce | THM | Dannce-THM | THM-Dannce | Calms2-Dannce | Calms2-THM |
|---|---|---|---|---|---|---|
| Baseline | 3.25 | 5.67 | 10.02 | 9.48 | 20.65 | 22.50 |
| RANSAC[36] | 2.65 | 4.96 | 9.66 | 8.01 | 18.39 | 17.50 |
| DeeplabCut[15] | 4.20 | 4.40 | 11.52 | 11.43 | 21.43 | 17.78 |
| GeneralTriang[10] | 2.56 | 4.65 | 8.56 | 8.06 | 18.55 | 16.50 |
| Sup3d*[9] | 2.34 | 3.45 | 7.33 | 7.15 | 15.50 | 14.46 |
| TR | 2.22 | 3.49 | 5.87 | 7.15 | 15.16 | 15.11 |
| TR+DD | - | - | 5.78 | 5.83 | 14.84 | 15.07 |
| TR+Sup3d* | **2.00** | **3.37** | **5.76** | **5.57** | **14.84** | **14.38** |

To demonstrate the "plug-and-play" ability of TR loss, we implement the experiments on four tasks by plugging the triangulation head and the confidence head into different 2D detectors (including PVT[37], SCNet[38], and MobileNet[39]). As shown in Table 3, all 2D detectors get lower MPJPE errors with TR loss. Especially for the cross-domain tasks (Dannce-THM and THM-Dannce), TR loss reduces the MPJPE by an average of 3.85mm (36%).

We also present the MPJPE results of each joint on the Dannce dataset in Appendix A.5. The results show that the nose and ears have lower errors than those of other points, while limb positions are difficult to estimate because of the self-occlusion, and body joint error is relatively large due to the lack of texture features.

Table 3: MPJPE results of different 2D detectors with and without TR

| | PVT | | | SCNet | | | MobileNet | | |
|---|---|---|---|---|---|---|---|---|---|
| | w/o TR | w/ TR | error | w/o TR | w/ TR | error | w/o TR | w/ TR | error |
| Dannce | 3.12 | 2.74 | **-0.38** | 3.27 | 2.48 | **-0.79** | 3.26 | 2.81 | **-0.45** |
| THM | 4.57 | 3.81 | **-0.76** | 4.47 | 3.49 | **-0.98** | 4.94 | 4.04 | **-0.90** |
| Dannce-THM | 12.87 | 7.96 | **-4.91** | 11.08 | 6.44 | **-4.64** | 11.15 | 5.57 | **-5.58** |
| THM-Dannce | 11.09 | 7.74 | **-3.35** | 8.49 | 5.48 | **-3.01** | 9.47 | 7.85 | **-1.62** |

## 4.4 Further analysis

**Data efficiency analysis** We compare the MEJPE of models trained with different data sizes and present the results in Figure 3 (a) - (b). Naive triangulation means triangulating directly based on the 2D estimates of the backbone. Supervised triangulation means training the total model with the ground-truth 3D labels. Our model is trained with TR loss. Figure 3 (a) shows that the MPJPE of all models decreases as the training data size increases. Our model performs better than other models in most cases especially when the training data size is small (5%-10%). The cross-subject results in Figure 3 (b) show that our self-supervised method can significantly improve the generalization ability of different individuals, but supervised triangulation fails.

**Sensitivity testing on the score threshold** To avoid the trivial solution, we set a threshold range to the confidence score. Figure 3 (c) shows the MPJPE results curves with different threshold ranges. We set the threshold range to be symmetric about 0.5 as shown on the x-axis, where "No" means model without score head and $(0.0, 1.0)$ means model without threshold. Models without a score head get degraded performance since they can not adjust the weight of each view in the triangulation process, while models without a score threshold get significantly large errors since TR loss to assign most views with a weight close to 0, in this way, these views have no contributions to triangulation. Except for these two extreme settings, our method obtains similar results because TR loss and confidence can be adaptively balanced during the training process.

**Ablation studies** Figure 3 (d) - (e) present the ablation experiment results. The baseline model and 3D supervised model share the same model structure as the "TR+confidence" model having the confidence head, while the "TR" model does not have the confidence head. The results show that TR

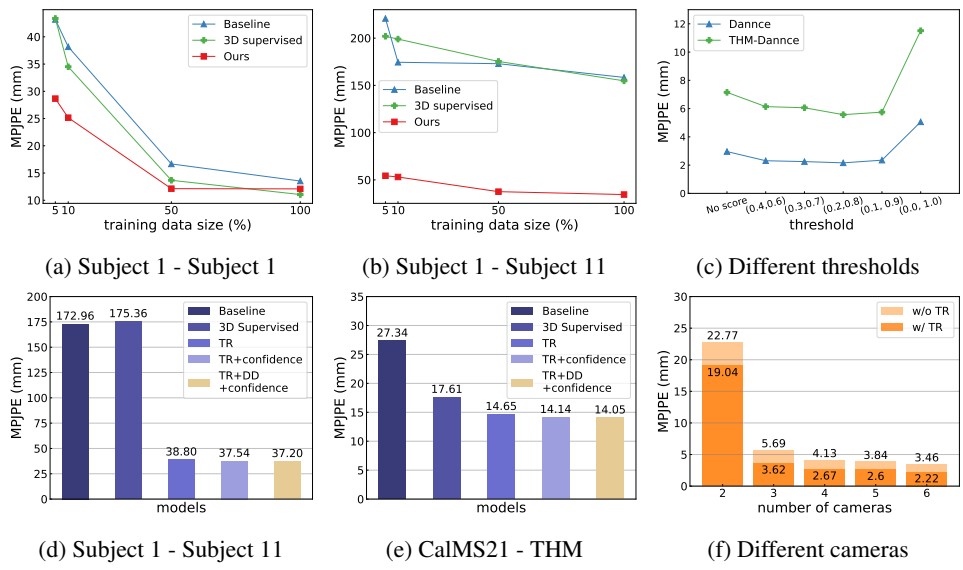

(a) Subject 1 - Subject 1  (b) Subject 1 - Subject 11  (c) Different thresholds

(d) Subject 1 - Subject 11  (e) CalMS21 - THM  (f) Different cameras

Figure 3: (a) - (b) Data efficiency analysis on Human3.6M dataset. (c) Sensitivity testing on the score threshold. (d) - (e) Ablation studies. The results show that TR loss is the most critical component and DD loss is optional. (f) Results on different numbers of cameras.

loss is the most critical component, yielding a 77.56% improvement on Subject 1-Subject 11 and a 46.41% improvement on the CalMS21-THM experiment compared to the baseline. The confidence head also provides a slight improvement due to its learnable weights selecting high-confidence views. The DD loss contributes the least to the performance.

**Results on different numbers of cameras** Figure 3 (f) shows the results on different numbers of cameras on the Dannce dataset. The error increases slightly as the number of cameras decreases from 6 to 3, and increases sharply when the number of cameras decreases to 2. However, the TR loss achieves consistent improvement for all camera numbers.

**Running time analysis** In the training process, our method is the same as typical deep learning models that compute the loss functions (including TR loss) only once per batch to update the weights. The TR loss does not require additional iterative optimization within a batch. Therefore, TR loss does not excessively increase the training time. During testing, our method also does not require iterative optimization. The experiments show that the average per-batch running time of the same model with and without TR loss on the Dannce dataset is 0.39s and 0.46s, respectively.

## 4.5 Qualitative results

Figure 4 shows qualitative examples of the Human3.6M dataset. The first four columns present the 2D estimates in different views generated by the 2D backbone, while the last column shows the 3D estimates of different methods. The 2D detector is pretrained in the same way as the "Subject 1 - Subject 11" experiment. As shown, the 2D backbone can not provide steady 2D estimates because of the domain discrepancy and limited training data. The naive triangulation and supervised method are incapable of correcting these errors (such as the right arm and right hip in view 2). However, our method corrects the failed 2D estimates in each view via global geometry consistency self-supervision, providing accurate 3D results.

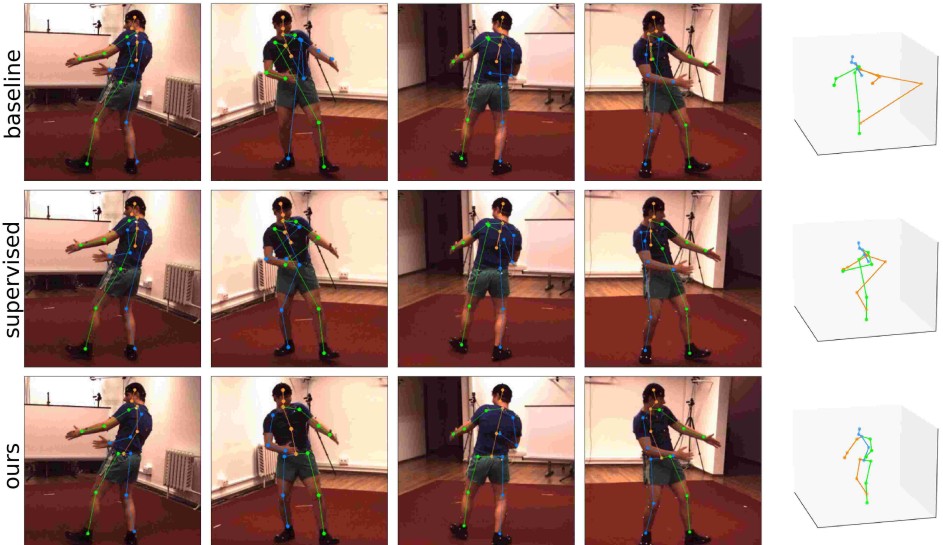

Figure 4: Qualitative results on Human3.6M dataset. Our method provides geometrically consistent 2D estimates in each view and an accurate 3D estimate.

The examples of mouse pose estimation in Figure 5 also show that our method obtains better 3D estimates. Due to space limitations, we only show three views and the 3D results here. All models in Figure 5 are pretrained on the THM training set. As shown in column 3 of Subfigure (a), only our method provides geometrically plausible 2D estimates for limbs and body. Moreover, only our method distinguishes the left paw and right paw in Subfigure (b). More qualitative results and failure examples can be found in Appendix A.6.

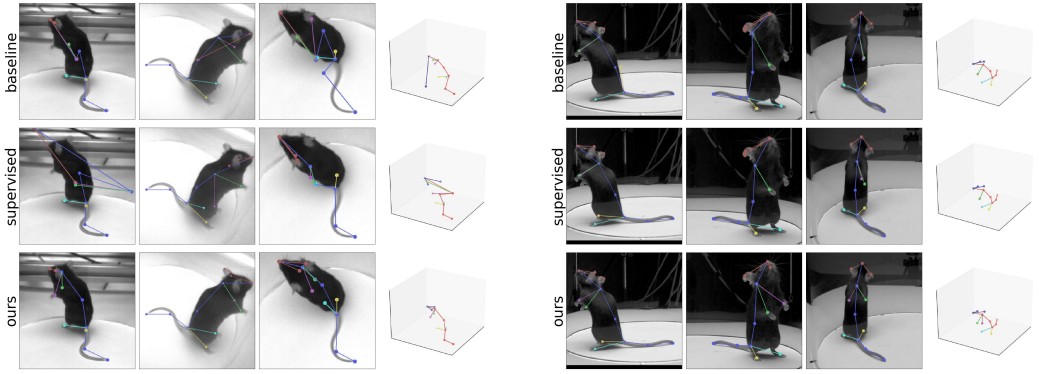

(a) THM dataset example        (b) Dannce dataset example

Figure 5: Qualitative results of mouse pose estimation. Our method obtains accurate results for the hard joint (body) and unseen joints (paws) in View 3.

## 5    Conclusion

In this paper, we present TR loss for multiview 3D human/animal pose estimation, which aims to minimize the sum of distances between 3D estimation and view rays. Its advantages contain (1) Self-supervised. It achieves global geometric consistency without 3D supervision and any reprojection processes. (2) Simple. It only minimizes the smallest singular value of the triangulation matrix and can be easily applied to various 2D backbones; (3) Effective and data efficient. It achieves state-of-the-art MPJPE performance for single-frame 3D pose estimation on the Human3.6M dataset and obtains competitive results with only 5% training data. Future work includes combining TR loss with end-to-end learnable association methods to generalize to multi-object pose estimation, or using TR loss to jointly optimize the camera poses and object poses for moving cameras.

## Acknowledge

We thank the reviewers for their constructive comments. This work was supported in part by Guoqiang Institute of Tsinghua University (No.2021GQG0001), NSFC (No.62171255), China Postdoctoral Science Foundation under Grant BX20220186, and Shuimu Tsinghua Scholar Program.

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
