# Supplementary Material of Triangulation Residual Loss for Data-efficient 3D Pose Estimation

## A Appendix

### A.1 Formulate the triangulation matrix $\widetilde{A}$

Triangulation aims to estimate a 3D point $\boldsymbol{X}^*$ based on the 2D measurement $\hat{\boldsymbol{x}}_c = (\hat{x}_c, \hat{y}_c, 1)$ in each view by solving equations $\boldsymbol{x}_c = \boldsymbol{P}_c \cdot \boldsymbol{X}^*$. The homogeneous method transforms the equations with a cross-product, as $\hat{\boldsymbol{x}}_c \times (P_c \boldsymbol{X}^*) = 0$, and gives three equations for each 2D measurement, of which two are linearly independent:

$$\begin{aligned}
\hat{x}_c \cdot (\boldsymbol{P}_{c,3}^T \cdot \boldsymbol{X}^*) - (\boldsymbol{P}_{c,1}^T \cdot \boldsymbol{X}^*) = 0 \\
\hat{y}_c \cdot (\boldsymbol{P}_{c,3}^T \cdot \boldsymbol{X}^*) - (\boldsymbol{P}_{c,2}^T \cdot \boldsymbol{X}^*) = 0 \\
\hat{x}_c \cdot (\boldsymbol{P}_{c,2}^T \cdot \boldsymbol{X}^*) - \hat{y}_c \cdot (\boldsymbol{P}_{c,1}^T \cdot \boldsymbol{X}^*) = 0
\end{aligned} \tag{1}$$

where $\boldsymbol{P}_i^T$ is the $i$th row of $\boldsymbol{P}$. Combining independent equations of all views gives an overdetermined linear system of equations as $\boldsymbol{A}\boldsymbol{X} = 0$ with

$$A = \begin{bmatrix}
\hat{x}_1 \cdot \boldsymbol{P}_{1,3}^T - \boldsymbol{P}_{1,1}^T \\
\hat{y}_1 \cdot \boldsymbol{P}_{1,3}^T - \boldsymbol{P}_{1,2}^T \\
\vdots \\
\hat{x}_C \cdot \boldsymbol{P}_{C,3}^T - \boldsymbol{P}_{C,1}^T \\
\hat{y}_C \cdot \boldsymbol{P}_{C,3}^T - \boldsymbol{P}_{C,2}^T
\end{bmatrix} \in \mathbb{R}^{2C \times 4}, \tag{2}$$

Taking the learnable weights into account, the triangulation matrix will be

$$\widetilde{A} = [w_1, w_1, ..., w_C, w_C]^T \odot A \tag{3}$$

where $[w_1, w_1, ..., w_C, w_C] \in \mathbb{R}^{2C}$.

### A.2 Relation of the geometric error and the algebraic error

The algebraic error Eq. (9) is approximate to geometric error Eq. (5), but the algebraic error can be solved linearly so it is more suitable for end-to-end training. A detailed discussion follows.

Given a point predicted by the 2D detector in $c$th view with homogeneous coordinates as $\hat{\boldsymbol{x}}(\hat{x}, \hat{y}, \hat{t})$. The triangulation problem is to solve overdetermined systems of equations $\hat{\boldsymbol{x}} = P_c \cdot \boldsymbol{X}^*$ where $c = 1, 2, ..., C$. We further define $\boldsymbol{x}^*(x^*, y^*, t^*)$ as the exact point projected from the estimated 3D point $\boldsymbol{X}^*(X^*, Y^*, Z^*, 1)$ using the projection matrix $P_c$ (i.e. $\boldsymbol{x}^* = P_c \cdot \boldsymbol{X}^*$). There are two ways to solve the triangulation problem: the inhomogeneous method (minimizing geometric error) and the homogeneous method (minimizing algebraic error). The geometric error is defined as the Euclidean distance between $\hat{\boldsymbol{x}}$ and $\boldsymbol{x}^*$ in inhomogeneous coordinates: $d_{\text{geo}}(\hat{\boldsymbol{x}}, \boldsymbol{x}^*) = ((\hat{x}/\hat{t} -$

$x^*/t^*)^2 + (\hat{y}/\hat{t} - y^*/t^*)^2)^{1/2}$, where $t^*$ is the depth of point $\boldsymbol{X}^*$ from the $c$th view, which usually varies across views. The algebraic error is the 2-norm of the algebraic error vector and writes $d_{\text{alg}}(\hat{\boldsymbol{x}}, \boldsymbol{x}^*) = ((\hat{y}t^* - \hat{t}y^*)^2 + (\hat{t}x^* - \hat{x}t^*))^{1/2}$. The homogeneous method finds the solution of minimizing $d_{\text{alg}}$ as the unit singular vector corresponding to the smallest singular value of A as shown in Eq.(6)-Eq.(9). Moreover, we always set $t = 1$, so $d_{\text{alg}} = d_{\text{geo}} * t^*$. By accumulating the errors from all the points from different views, $d_{\text{alg}}$ is not proportional to $d_{\text{geo}}$ because $t^*$s from different views are unequal. Minimizing the algebraic error is an approximation of minimizing the geometric error, but it can be solved linearly and is more suitable for end-to-end training. See Chapters 4.1 and 12.1 in "Multiple view geometry in computer vision" (Hartley, 2003) for details.

### A.3   Pseudocode for computing TR loss

Algorithm 1 presents the procedure to compute TR loss for a joint in $C$ synchronized multiview images.

---
**Algorithm 1:** compute TR loss

---
**Input:** 1. synchronized multiview images $\boldsymbol{I}_c\ (c = 1, 2, ..., C)$;
       2. projection matrix of cameras $\boldsymbol{P}_c\ (c = 1, 2, ..., C)$.
**Parameters:**1. a pretrained 2D detector with backbone $\boldsymbol{\Phi}$ and heatmap head $\boldsymbol{h}$;
        2. a randomly initialized confidence head $\boldsymbol{f}$.
**Output:** 1. TR loss, $\mathcal{L}^{res}$;
       2. updated model parameters.
1 propagate images $\boldsymbol{I}_c$s forward through the total model;
2 compose the triangulation matrix $\widetilde{\boldsymbol{A}}$ via Eq. (3) based on outputs of the model;
3 perform SVD on $\widetilde{\boldsymbol{A}}$;
4 $\mathcal{L}^{res} \leftarrow$ the smallest singular value of $\widetilde{\boldsymbol{A}}$;
5 update the model by minimizing $\mathcal{L}^{res}$.
6 **return** $\mathcal{L}^{res}$; updated $\boldsymbol{\Phi}$, $\boldsymbol{h}$ and $\boldsymbol{f}$.

---

### A.4   Mouse skeleton definition

The original CalMS21 dataset contains 30,000 frames of two interacting mice (a black mouse and a white mouse) performing resident-intruder assay in a homecage. We only consider the single object pose estimation in this paper, so we regard the black mouse as the object and the white mouse as the background. Images in the CalMS21 dataset are captured by a top-view camera and a front-view camera. Since the self-occlusion, 7 joints are annotated for the front view and 11 joints are annotated for the top view. We select 9 joints to evaluate methods, as shown in Figure 1 (a) and Table 1. Some images in CalMS21 are not high quality due to insufficient lighting and occasion by bedding in the mouse cage. Therefore, we only keep parts of high-quality images as training data in this paper. To select the high-quality images, we train and test a 2D key point detection model on the entire dataset, and then select the images with high accuracy. Finally, 767 images from the front view and 12 from the top view are used as training data in the CalMS21-Dannce and CalMS21-THM experiments.

The original Dannce dataset annotates 22 joints, from which we select 12 joints in this paper, as shown in Figure 1 (b) and Table 2. We follow the same 12-joint definition and annotate our THMouse dataset. We use the 9-joint definition in the CalMS21-Dannce and CalMS21-THM, and the 12-joint definition in the THM-Dannce and Dannce-THM experiments.

Table 1: Joint definition of the CalMS21 dataset

| Index | 0 | 1 | 2 | 3 | 4 | 5 | 6 | 7 | 8 |
|---|---|---|---|---|---|---|---|---|---|
| **Joint** | left ear | right ear | nose | neck | tail_1 | left paw | right paw | left foot | right foot |

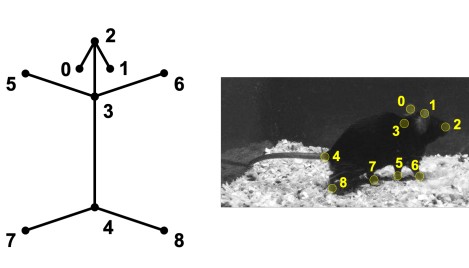

(a) 9 joints in CalMS21 dataset

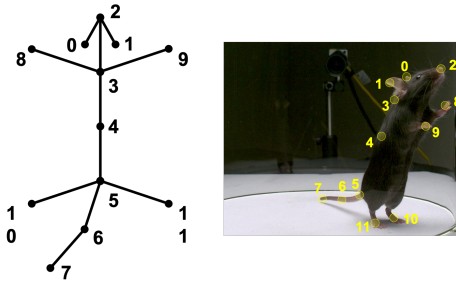

(b) 12 joints in Dannce and THmouse datasets

Figure 1: Mouse skeleton definition

Table 2: Joint definition of the Dannce and the THmouse dataset

| Index | 0 | 1 | 2 | 3 | 4 | 5 |
|---|---|---|---|---|---|---|
| Joint | left ear | right ear | nose | neck | body | tail_1 |
| Index | 6 | 7 | 8 | 9 | 10 | 11 |
| Joint | tail_2 | tail_3 | left paw | right paw | left foot | right foot |

## A.5 MPJPE result on each joint on Dannce dataset

Table 3 shows the MPJPE error on the Dannce dataset. The results show that the nose and ears have lower errors than those of other points, while limb positions are difficult to estimate because of the self-occlusion, and body joint error is relatively large due to the lack of texture features.

Table 3: MPJPE result of each joint on Dannce dataset. The unit is mm.

| | Ear_L | Ear_R | Nose | Neck | Body | Tail1 | Tail2 | Tail3 | Paw_L | Paw_R | Foot_L | Foot_R | Average |
|---|---|---|---|---|---|---|---|---|---|---|---|---|---|
| Baseline | 1.27 | 1.37 | 1.56 | 2.85 | 3.63 | 2.78 | 3.39 | 1.54 | 4.67 | 5.99 | 3.94 | 3.04 | 3.04 |
| Ransac | 1.04 | 1.23 | 1.80 | 2.50 | 3.07 | 3.04 | 3.36 | 2.35 | 3.98 | 3.56 | 2.95 | 2.43 | 2.61 |
| DeeplabCut | 2.84 | 2.51 | 2.21 | 3.93 | 4.10 | 3.98 | 4.22 | 3.47 | 5.57 | 5.31 | 5.44 | 4.68 | 4.02 |
| GeneraliTriang | 0.91 | 1.32 | 1.79 | 2.80 | **2.90** | 3.06 | 3.74 | 2.45 | 3.36 | 3.69 | 2.50 | **2.28** | 2.56 |
| Sup3d* | 0.81 | 0.95 | 1.08 | 2.45 | 3.14 | 1.20 | 4.60 | 2.38 | 2.18 | 2.86 | **2.37** | 4.14 | 2.35 |
| TR+Sup2d | 1.02 | 1.02 | 2.13 | 2.80 | 3.50 | 1.43 | **2.84** | **1.56** | 2.83 | 2.23 | 2.55 | 2.75 | 2.22 |
| TR+Sup3d* | **0.69** | **0.72** | **1.05** | **2.25** | 3.20 | **1.00** | 4.58 | 1.82 | **2.03** | **1.94** | **2.37** | 2.39 | **2.00** |

## A.6 More qualitative results

More good examples are shown in Figure 2-4.

Some failed examples are shown in Figure 5.

A video example is available at https://www.youtube.com/watch?v=CjuV5qEfiFg.

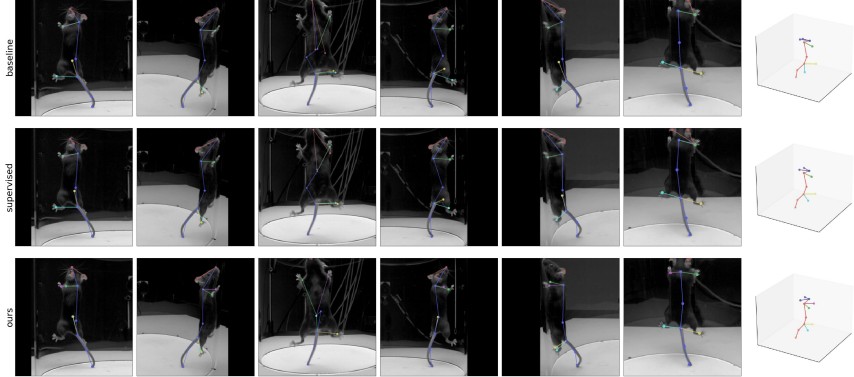

Figure 2: More qualitative results on the Dannce dataset.

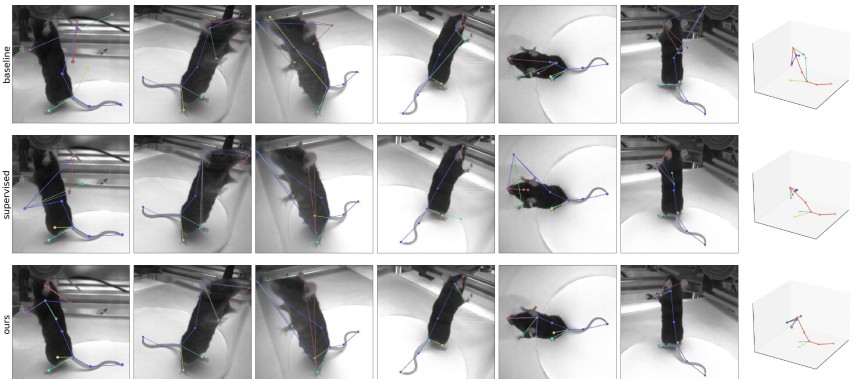

Figure 3: More qualitative results on the Thmouse dataset.

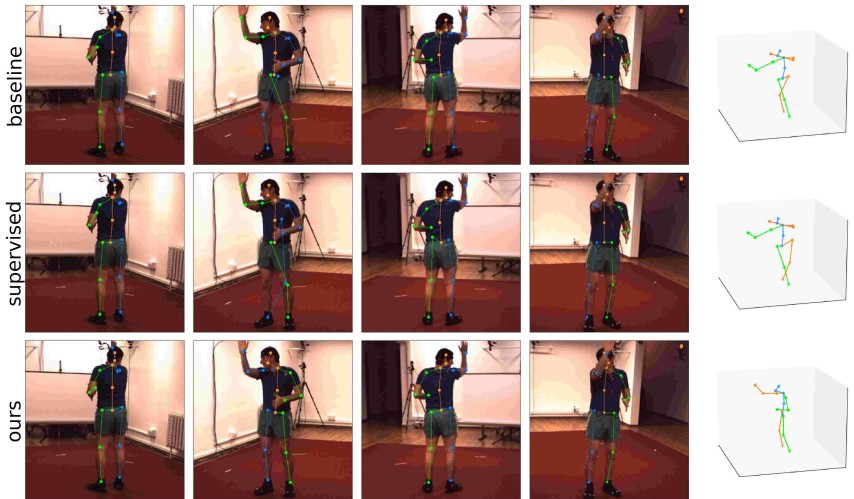

Figure 4: More qualitative results on Human3.6M dataset, where models are trained on Subject 1 but tested on Subject 11.

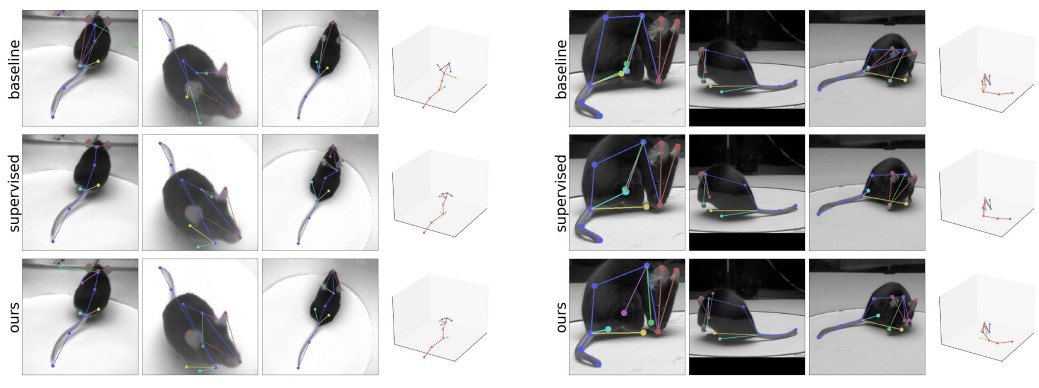

(a) THM dataset example     (b) Dannce dataset example

Figure 5: Failed examples of mouse pose estimation, where the lambs are not detected correctly due to the self-occlusion.