# OpenReview forum: "Triangulation Residual Loss for Data-efficient 3D Pose Estimation"
_NeurIPS.cc/2023/Conference — NeurIPS 2023 poster_

### Official Review · Reviewer_iymk · 2023-07-04

**Soundness:** 3 good
**Presentation:** 3 good
**Contribution:** 3 good
**Rating:** 5
**Confidence:** 2

**Summary:**

This paper focuses on the task of multi-view 3D human/animal pose estimation, and proposes Triangulation Residual Loss, termed as TR Loss, to constraint multi-view geometry consistency in a self-supervised manner. TR Loss is used to minimize the distances between the predicted 3D point to the view rays, and simply implemented as minimizing the smallest singular value of the triangulation matrix. Experiments verify the effectiveness of TR Loss on both laboratory mouse pose estimation and human pose estimation.

**Strengths:**

- TR Loss is simple yet effective, which self-supervises the multi-view geometry consistency by minimizing the smallest singular value of the triangulation matrix without 3D annotations or heavy computation of reprojection.

- A new THmouse dataset is constructed to help build a benchmark for laboratory mouse pose estimation.

- SOTA MPJPE results on Human3.6M dataset are achieved, which verify the effectiveness of TR Loss.


**Weaknesses:**

- In Fig. 1, there are three groups of methods discussed; however, the quantitative comparisons with those of the second group (cf. Fig. 1b), especially in terms of accuracy and efficiency, are not presented in the experiments.

- For the newly built benchmark of laboratory mouse 3D pose estimation, experiments in Sec. 4.2 seem to be ablation studies. No SOTA methods are directly applied on this benchmark. It’s suggested to provide the results of some recent methods.


**Questions:**

- In Line158, the weights are scaled to (0.4, 0.6). Does it limit the effect of the predicted confidence? Why the superparameters (0.4 and 0.6) are set? Ablation studies are needed.

- In Table 2, the best results are not bolded.

- In Table 3, it seems that the result of MTFT-Transformer is not consistent with that in their paper. And there are some methods listed in their paper and not compared in this paper, e.g, [1] and [2], which achieve better results. For different ratios of training data (5% and 100%), what are the ratios of unlabeled data?

- In Fig. 3(b), why TR Loss has stronger generalization ability than 3D supervised loss?

Reference:

[1] AdaFuse: Adaptive Multiview Fusion for Accurate Human Pose Estimation in the Wild.

[2] Fusing Wearable IMUs with Multi-View Images for Human Pose Estimation: A Geometric Approach.

**Limitations:**

The authors do not discuss the limitations and potential negative societal impacts of their work in the paper.

---

> ### Author Rebuttal · Authors · 2023-08-02
>
> > Q1: In Fig. 1, there are three groups of methods discussed; however, the quantitative comparisons with those of the second group (cf. Fig. 1b), especially in terms of accuracy and efficiency, are not presented in the experiments.
>
> R1: We chose GeneralTriang[10] as a comparison method here. As the results in the result table in   R2 shown, the GeneralTriang performs worse than our method. It is also less efficient than our method. The average per-batch running time of the baseline model, our method, and GeneralTriang are 0.389s, 0.463s, and 2.05s. GeneralTriang takes a longer time since it needs to select view subsets and update the 3D hypotheses iteratively in each batch. Our method just calculates one more loss function than baseline so the computational time complexity is nearly the same.
>
>
>
> > Q2: For the newly built benchmark of laboratory mouse 3D pose estimation, experiments in Sec. 4.2 seem to be ablation studies. No SOTA methods are directly applied on this benchmark. It’s suggested to provide the results of some recent methods.
>
> R2: We add the DeepLabCut and GeneralTriang[10] as the SotA comparison methods. We chose these methods because DeepLabCut is the most popular animal pose estimation method, and GeneralTriang is the latest SotA pose estimation method proposed in CVPR 2022. The results are shown in the following table.
>
> |            | DeepLabCut | GeneralTriang | Ours |
> |:----------:|:----------:|:-------------:|:----:|
> |   Dannce   |    4.20    |      4.15     | 3.54 |
> | THM-Dannce |    11.43   |      7.29     | 5.18 |
>
>
> In addition, We also plug our TR loss onto different 2D detectors. including PVT, SCNet, and MobileNetV2 (referring R1 to reviwer y7FT). The MPJPE errors in the following table show that our modules and loss functions provide consistent and improved results on all 2D detectors.
>
>
>
> > Q3: In Line158, the weights are scaled to (0.4, 0.6). Does it limit the effect of the predicted confidence? Why the superparameters (0.4 and 0.6) are set? Ablation studies are needed.
>
> R3: Learnable confidence mitigates the negative effects of inaccurate 2D estimates to triangulation. But in turn, it also affects the updating value of inaccurate 2D estimates. As discussed in L153, if the weights of a view are too small, then its 2D estimate is hardly updated. Theoretically, as long as the weights of each view are not all equal or differ by orders of magnitude (one with 1e-4, one with 9.999), TR loss can obtain a balance between mitigating and updating the inaccurate 2D estimates. We supplemented the sensitivity experiments with confidence thresholds, which showed that setting the parameters at (0.4, 0.6)-(0.1, 0.9) gives approximate results that are significantly better than without confidence and (0,1).
>
> |                | without confidence | (0.4, 0.6) | (0.3, 0.7) | (0.2, 0.8) | (0.1, 0.9) | (0,1) |
> | :------------: | :----------------: | :--------: | :--------: | :--------: | :--------: | :---: |
> | Dannce dataset |        4.54        |    3.61    |    3.74    |    3.60    |    3.90    | 5.89  |
> |   THM-Dannce   |        8.11        |    6.21    |    6.52    |    6.72    |    6.62    | 29.42 |
>
> > Q4: In Table 2, the best results are not bolded.
>
> R4: We thank the reviewers for the careful review and we will update Table 2 in the revised paper.
>
> > Q5: In Table 3, it seems that the result of MTFT-Transformer is not consistent with that in their paper. And there are some methods listed in their paper and not compared in this paper, e.g, [1] and [2], which achieve better results. For different ratios of training data (5% and 100%), what are the ratios of unlabeled data?
>
> R5: Sorry for the mistake, we corrected the performance of the MTFT-Transformer to 27.46mm according to TABLE 9 of the MTFT-Transformer paper. We also checked the results of other compared methods.
>
> References [1] and [2] are both outstanding works in human pose estimation, AdaFuse [1] fuses the features of two corresponding points in two different views. ORPSM [2] integrates the IMU signal and multiview images to improve both 2D and 3D pose estimation. We will discuss the difference between them and our manuscript. However, AdaFuse used the MPII dataset as additional data, while ORPSM used the IMU data.
>
> For the 100\% training data, we compute the TR loss on 100\% training data with 2D label and 1\% test data with only image (without 2D/3D supervision). For the 5\% training data, we compute the TR loss on 5\% training data with 2D label and other 95\% training data with only image (without 2D/3D supervision). We used such a setting because we try to demonstrate TR loss can be used on unlabeled data.  We will add the experiments with only training data.
>
> > Q6: In Fig. 3(b), why TR Loss has stronger generalization ability than 3D supervised loss?
>
> R6: The 3D supervised loss fits the 3D ground truth mainly by weighting the different views, so the 3D supervised loss may be small but the 2D estimates are erroneous. The  3D supervised loss does not have a direct cue to improve the 2D detector. Instead, the TR loss successfully enforces the geometrical consistency by minimizing the sum of distances between 3D estimation and view rays. Therefore, it is a reasonable result that TR loss demonstrates better generalization ability in cross-dataset experiments.
>
> > Q7: Limitations and societal impacts.
>
> R7: The limitation of our method is that it requires a 2D detector in giving a relatively accurate initial estimate. This is also a common limitation of 3D pose estimation without 3D supervision. We present some failure cases in the appendix.
> In this paper, we propose a solution for non-3d supervision pose estimation for both humans and animals, which has potential applications for both human and animal behavior analysis. As mentioned by other reviewers, our method and results have no negative impact on animal conservation and welfare, human health and safety, and social ethics.

---

### Official Review · Reviewer_y7FT · 2023-07-05

**Soundness:** 2 fair
**Presentation:** 2 fair
**Contribution:** 2 fair
**Rating:** 5
**Confidence:** 3

**Summary:**

The authors present an approach for 3D pose estimation that leverages multi-view stereo cues. A weak 2D keypoint detector is refined using an unsupervised triangulation consistency from posed cameras (TR loss) which iteratively minimizes triangulation discrepencies. The method is experimentally evaluated on an animal dataset of mice and Human3.6M.

**Strengths:**

* S1. Multi-view consistency as refiner. The idea to use multi-view consistency to refine a (potentially weak) 2D keypoint detector with a cross-view loss is interesting.
* S2. Simple idea with potential. The use of unsupervised consistency loss can lift simpler 2D labels into 3D where annotation is more cumbersome.

**Weaknesses:**

* W1. Missing Baselines / Backup Experiments. In details:
   * L16 claims a "plug-and-play module that enables data-efficient training of all 2D keypoint detectors". However, the 2D keypoint detectors are not changed in experiments.
   * The related work talks about [11,12,13,14] as previous MVS setups - and L91 speaks about "local pariwise consistency", however, this can be globally optimized. Would it be possible to include a comparison against commonly used reconstruction methods / optimizations such as the one from COLMAP? No competitor baselines are used for comparison such as [11,12,13,14, ...], but only self-made baselines. Why?
   * The experiments on H3.6M are compared against 3D supervision. The idea of the paper, however, is conceptual. How would these methods/pipelines improve with the TR loss?
* W2. Quesitonable robustness. Given the iterative nature of the MVS triangulation consistency, it is unclear to me how an outlier measurment would affect the procedure. Where is the robustness to single outliers in the process?
* W3. Missing mathematical rigorosity. There are a couple of unclarities / mathematical details that should be specified such as:
   * Eqn (5): It is unclear whihc norm is used? L2? Depending on this, the equivalence of (5) and (6) migh not be given. Would it help to add a discussion to the paper about the minimization of a geometric vs. an algebraic error?
   * Considerations for non-degenerate / degenerate constraints for eqn (6) are missing. E.g. rank considerations
   * Around eqn (8). \sigma_i, u_i, v_i should be defined
   * L149f sigma not necessary in strict order >= should be used
   * L152 / eqn (9): What happens if the smallest eigenvalue is not unique?
   * Eqn (11), L168ff z with hat is missing a definition
* W4. Minor grammatical errors etc. / typos
   * L53: colloquial: "doesn't"
   * L87: captial letters "All"
   * L95: \sigma_4 not defined
   * L102/L104/L112/L121 notation arrow is typically used to define the mapping (not the sets)
   * L114: missing space "(1)and"
   * Template: NeurIPS22
   * Eqn.(1): Consistency on the naming (e.g. not italic)
   * L117: "indices"
   * L123: "objective"?
   * L131: norm not defined (L1 or L2 used?)
   * L135: "Eq.(6)" missing space
   * L145: norm definition is missing
   * Eqn (7): norm not defined (also in eqn (8,9,10))
   * Colloquial (L209 / L211): doens't, can't
   * L261: doen't
   * L270: can't

Most critical points have been addressed during the rebuttal!

**Questions:**

* Q1. Given the iterative nature of the MVS triangulation consistency, it is unclear to me how an outlier measurment would affect the procedure. Where is the robustness to single outliers in the process?
* Q2. What is the reason for the choice of solely self-made baselines? Why are no MVS methods compared? How would other methods work if the TR loss is used (e.g. additionally)?
* Q3. What would be the influence of camera pose error on the result?
* Q4. Are all other baselines (H3.6M) trained with the same data amount (100%)? Are they most recent SotA?


**Limitations:**

Failure cases are discussed in appendix.

---

> ### Author Rebuttal · Authors · 2023-08-02
>
> The authors thank the reviewer for constructive and helpful feedback. To address your questions and concerns:
>
> > Q1: L16 claims a "plug-and-play module ... However, the 2D keypoint detectors are not changed in experiments.
>
> R1: To demonstrate the "plug-and-play" ability, we complemented the experiments on the Dannce dataset by plugging our triangulation head and confidence head on different 2D detectors (including PVT [r1], SCNet [r2], and MobileNetV2 [r3]). In the following table, "baseline" triangulates the 3D estimates based on the 2D results of each 2D detector, "baseline+TR" is finetuned with our TR loss. The MPJPE errors show that our modules and loss functions provide consistent improvements on all 2D detectors.
>
> |    Dannce     | PVT  | SCNet | MobileNet |
> | :-----------: | :--: | :---: | :-------: |
> |   Baseline    | 4.11 | 3.85  |   5.78    |
> | Baseline + TR | 2.94 | 2.86  |   4.33    |
>
> | THM-Dannce | PVT  | SCNet | MobileNet |
> | :--------: | :--: | :---: | :-------: |
> |  Baseline  | 9.44 | 9.29  |   11.13   |
> |   Baseline +  TR     | 6.25 | 6.32  |   7.22    |
>
> [r1] Pyramid Vision Transformer: A Versatile Backbone for Dense Prediction without Convolutions, ICCV, 2021
>
> [r2] Improving Convolutional Networks With Self-Calibrated Convolutions, CVPR, 2020
>
> [r3] MobileNetV2: Inverted Residuals and Linear Bottlenecks, CVPR, 2018.
>
>
>
> > Q2: The related work ..., such as the one from COLMAP? No competitor baselines are used for comparison such as [11,12,13,14, ...], but only self-made baselines. Why?
>
> R2: We add the DeepLabCut[r4] and GeneralTriang[10] as the SotA comparison methods. We chose these methods because DeepLabCut is the most popular animal pose estimation method, and GeneralTriang is the latest SotA pose estimation method proposed in CVPR 2022. GeneralTriang is also based on triangulation. The results are shown in the tables in R1.
>
> [r4] Using DeepLabCut for 3D markerless pose estimation across species and behaviors, Nature Protocols, 2019.
>
> |            | DeepLabCut | GeneralTriang | Ours |
> |:----------:|:----------:|:-------------:|:----:|
> |   Dannce   |    4.20    |      4.15     | 3.54 |
> | THM-Dannce |    11.43   |      7.29     |  5.18    |
>
> We have not directly compared with COLMAP since its typical implementation of the triangulation is nearly the same as the "baseline" and "RANSAC" in Table.1 and Table. 2. Specifically, i) ``TriangulateMultiViewPoint`` is the core function for COLMAP triangulation and it uses DLT (the same as our baseline). ii)  According to its document, "a 3D similarity transformation will be estimated with a RANSAC estimator to be robust to potential outliers in the data.", and we have compared with RANSAC in Table.1. We will clarify this in the revision.
>
> > Q3: The experiments on H3.6M ..., How would these methods/pipelines improve with the TR loss?
>
> R3: Validating the effectiveness of the TR Loss under different setups is important. Thus, we have validated the effectiveness of the TR Loss even with 3D supervision in Table.1 on the mouse dataset. Moreover, the validity of the method is also demonstrated when 3D GT labels are unavailable (referring to R1). We will add additional validations on human datasets in the revision to make it more thorough.
>
>
> > Q4:Quesitonable robustness.
>
> R4: In the classical MVS triangulation,  2D outliers are excluded based on the reprojection errors during iterative optimization. However, 2D estimates in each view are not corrected throughout the process (including the outlier). Differently, our method updates the 2D estimates in an end-to-end training framework, and the TR loss forces the 2D outliers in each view to gradually converge to the positions with 3D consistency. (As mentioned by Reviewer ZQc8: "The iterative triangulation residual is a good formulation to realign the predicted heatmap locations to the accurate point using multi-view consistency"). We will release our code for reproducibility.
>
> > Q5: Missing mathematical rigorosity.
>
> R5: [1] Eqn(5) uses L2 norm. In fact, Eqn(5) and Eqn(6) are not fully equivalent, and we will tune the claim. Please refer to our answer "R2" to reviewer 4Fs8 for detailed discussion.
>
> [2]  To avoid degenerate cases where A is not a full-rank matrix, we simply ignore the TR loss where the condition number of A is too large.
>
> [3] The definition of \sigma_i, u_i, v_i are presented in L149-L150 actually. We find that it does not follow Eqn (8) closely, which is confusing. We will clarify it.
>
> [4] In L.149, we will use >=.
>
> [5] We simply use the last eigenvalue of SVD. In the used datasets, cameras are arranged reasonably and equal samllest eigenvalue is hard to observe.
>
> [6] We will clarify z hat.
>
> > Q6: Minor grammatical errors etc. / typos
>
> R6: We will go through the paper carefully with a native speaker.
>
> > Q7: Given the iterative ... Where is the robustness to single outliers in the process?
>
> R7: Please refer to R4.
>
> > Q8: What is the reason for the choice of solely self-made baselines? ...
>
> R8: Please refer to R1 and R2.
>
> > Q9: What would be the influence of camera pose error on the result?
>
> R9: Currently, we assume the camera poses are correct as other related methods. However, the differentiable nature of TR loss may have the potential to correct erroneous camera poses.
>
> > Q10: Are all other baselines (H3.6M) trained with the same data amount (100%)? Are they most recent SotA?
>
> R10: Yes. All the other baselines in Table 3 are trained with 100% data. They are the SotA under the setting of single-frame, multi-view 3D human pose estimation on the Human3.6M dataset without using additional training data. Notice that some papers achieved better results in different settings, for example, STCFormer (CVPR2023) and DiffPose (CVPR2023) used the ground-truth 2D pose as input, Token-Pruned Pose Transformer (ECCV2022), AdaFuse(IJCV2021), TesseTrack(CVPR2021) used the additional training data. We exclude these methods for fair comparision.

---

> > ### Comment · Reviewer_y7FT · 2023-08-16
> >
> > Many thanks for the additional experimental evidence and the correction of the mathematical flaws!
> > I would encourage the authors to explicitly include a statement such as the one from ZQc8 or the answer to Q4 to make this point clear.
> > While I was very unsure whether these things are possible within the rebuttal period, I am quite more positive about the paper now given that the commonly raised points (see your general answer 1,2,3,4) are adopted in the paper.
> > I therefore provisionally raise my rating to a "boarderline accept" trusting in the authors to make these changes.

---

> > > ### Author Response · Authors · 2023-08-17
> > > **Thanks for Reviewer y7FT for the responses**
> > >
> > > We greatly appreciate the reviewer's positive response to our revision and are glad to see the score change. We will present all changes of our rebuttal in the final paper. Thanks again for the patient review and constructive comments.

---

> > > > ### Comment · Reviewer_y7FT · 2023-08-20
> > > >
> > > > The additional video provided the TR-loss analysis is great. Please make sure it is included/linked in the paper.

---

> > > > > ### Author Response · Authors · 2023-08-20
> > > > >
> > > > > Many thanks for your kind reminder, we will link it in our final paper.

---

### Official Review · Reviewer_3C6c · 2023-07-05

**Soundness:** 3 good
**Presentation:** 3 good
**Contribution:** 3 good
**Rating:** 6
**Confidence:** 5

**Summary:**

This paper proposes to perform 3D pose estimation from multi-view RGB images. The key contribution is to develop a new loss function to enable effective training with only 2D pose supervision. This loss function is intended to iteratively optimize the geometric consistency between multi-view rays of each keypoints. The goal is to minimize the distance between the initial 3D estimate and multi-view rays to converge to a stable 3D position. The performance has been evaluated on one 3D human pose dataset, Human3.6M, and multiple mouse pose datasets.

**Strengths:**

1. Impressive quantitative results are achieved on multiple benchmarks. Extensive ablation studies testify the effectiveness of key designs.
2. The idea of developing a iterative loss to alleviate the in-consistency between 2D pose supervision is awesome.
3. The paper is well written and easy to follow.

**Weaknesses:**

1. Reproducibility. The results obtained in this paper is very attractive.  The code may not be released. The 5-lines implementation details at Sec. 4.1 clearly are not enough for reproducing the results. There is no guarantee for the reproducibility.
2. Limited qualitative results are provided to show the performance. More visual results, such as a video, would be very helpful.
3. Limitations are not discussed.

**Questions:**

Is there any way to ensure the reproducibility of the results in the paper?
How does the proposed method perform in the wild?

**Limitations:**

There are some obvious limitations, which have not been discussed in the paper.
For instance, the performance is heavily relying on the initial 2D pose estimation. While the in-the-wild performance of this part has not been validated.

---

> ### Author Rebuttal · Authors · 2023-08-02
>
> We appreciate the reviewers' careful review and constructive comments. To address your questions and concerns:
>
> > Q1: Reproducibility. The results obtained in this paper is very attractive. The code may not be released. The 5-lines implementation details at Sec. 4.1 clearly are not enough for reproducing the results. There is no guarantee for the reproducibility.
>
> R1: We will open-source our code and dataset if our paper can be accepted.** We will also improve the implementation details. Model details:  We apply HRnet as the backbone, build the heatmap head with a convolution layer with a 3x3 kernel size, and build the confidence head with two convolution layers followed by three linear layers [512, 256, num_joint] with a sigmoid activation function. The domain discriminator is an average pooling layer followed by three linear layers [512, 256, num_joint] with a sigmoid activation function. Training details: All models are trained on 1 NVIDIA 3090 GPU and an Intel  i7-11700 CPU with the Adam optimizer and an initial learning rate of 1e−5. We will also add more specific implementation details for each dataset in the appendix.
>
> > Q2: Limited qualitative results are provided to show the performance. More visual results, such as a video, would be very helpful.
>
> R2: We add a video comparing qualitative results w/o TR loss with w/ TR loss on Dannce dataset. TR loss largely improves the visual quality by removing obvious errors such as floating legs/paws or falsely overlaped front paws.
>
> > Q3: Limitations are not discussed.
>
> R3: Our method's limitation is that it relies on a pretrained 2D detector, which is a common drawback in 3D pose estimation without 3D supervision, especially for triangulation-based methods. We have shown some failure cases in the appendix.
>
> > Q4: Is there any way to ensure the reproducibility of the results in the paper? How does the proposed method perform in the wild?
>
> R4: We will release our code and datasets on GitHub as long as our paper is accepted.  As can be expected, our method can be applied in the wild. The 2D keypoint detector and 2D dataset are well-established, especially the human keypoint detector. As the 2D results shown in existing papers and the MMPose package, the 2D detector trained on the COCO dataset shows great in-the-wild performance, which can provide enough initial values.
>
> > Q5: There are some obvious limitations, which have not been discussed in the paper. For instance, the performance is heavily relying on the initial 2D pose estimation. While the in-the-wild performance of this part has not been validated.
>
> R5: Similar to common triangulation-based methods, our method also relies on the initial 2D pose estimation. However, most of the existing triangulation-based methods can not update inaccurate 2D estimates but only reduce their negative impact. On the contrary, our method can optimize the 2D estimates unsupervised. Thanks to the reviewer for the suggestions, we will make this more clear and evaluate our method on the in-the-wild datasets.

---

> > ### Comment · Reviewer_3C6c · 2023-08-19
> > **Performance shown in the video helps, I have upgraded to W.A.**
> >
> > Thanks for providing such an impressive video, which makes the performance be more convincing.
> > Providing code would also help the reproducibility.
> >
> > Therefore, most of my concerns have been resolved.
> > I have upgraded rating to W.A.

---

> > > ### Author Response · Authors · 2023-08-19
> > >
> > > Many thanks for your patient review and constructive comments. We will open-source our code upon acceptance.

---

### Official Review · Reviewer_ZQc8 · 2023-07-06

**Soundness:** 3 good
**Presentation:** 4 excellent
**Contribution:** 3 good
**Rating:** 6
**Confidence:** 5

**Summary:**

The paper proposes an triangular residual loss to optimize for the optimization of the 3D locations of the pose estimation. the iterative optimization framework address the problem of the erroneous predictions of the keypoints using learning based methods. The results are shown on multile pose estimation datasets with different objects like humans and mouses.

**Strengths:**

The paper has addressed an fundamental problem in triangulation i.e. is uncertain estimate of the keypoints in 2D and how to optimize for the 3D location given that the 2D locations have uncertainty.

The iterative triangulation residual is a good formulation to realign the predicted heatmap locations to the accurate point using multi-view consistency.

The method shows substantial result improvement on multiple datasets. Further they show reasonable improvements using much fewer images.

Such formulation is easily generalizable and analysis has been provided for the same.

**Weaknesses:**

Compared to the previous methods using a single loss over the 3D triangulation, in the current framework using iterative formulation causes additional compute and time to optimize. Analyzing the tradeoff in time vs accuracy compared to baselines will be helpful.

Experiments on additional datasets and with different objects should be explored .i.e. using panoptic studio or more datasets with different animals like monkey might show more benefits of the current formulation.

Although the method works with 5-6 camera views. analyzing what happens with fewer than that like 2-4 should be analyzed.



**Questions:**

what is the training time of the current formulation?

What happens if the estimated keypoint from one of the views is erroneous?



**Limitations:**

Limitations and societal impact have not been discussed.

---

> ### Author Rebuttal · Authors · 2023-08-09
>
> We thank the reviewer for the careful and constructive review. To address your questions and concerns:
> > Q1: Compared to the previous methods using a single loss over the 3D triangulation, in the current framework using iterative formulation causes additional compute and time to optimize. Analyzing the tradeoff in time vs accuracy compared to baselines will be helpful.
>
> R1: In the training process, our method is the same as typical deep learning models that compute the loss functions (including TR loss) only once per batch to update the weights. The TR loss does not require additional iterative optimization within a batch. So TR loss does not excessively increase the training time. During testing, our method also does not require iterative optimization.
> Regarding the training efficiency, we empirically compare the training time of our model with the baseline model. Both models are running under the same conditions (same batch size, same optimizer) on the same device (an NVIDIA 3090 GPU and Intel  i7-11700 CPU). The average per-batch running time of the baseline model and our model on the Dannce dataset are 0.389s and 0.463s. Therefore, the training time efficiency of our method is (0.463-0.389)/0.389=19\% lower than baseline, but the performance significantly improved by (5.86-3.61)/3.61=62\%. We will add a more systematic discussion and analysis of training time in the revised paper.
>
> > Q2: Experiments on additional datasets and with different objects should be explored. i.e. using panoptic studio or more datasets with different animals like monkey might show more benefits of the current formulation.
>
> R2: We thank the reviewers for the constructive comments. We apologize for not being able to provide the experimental results within the short deadline of the rebuttal. We will add more quantitative and qualitative experiments on different subjects in the revision.
>
> > Q3: Although the method works with 5-6 camera views. analyzing what happens with fewer than that like 2-4 should be analyzed.
>
> R3: We evaluate the results of different numbers of camera views on the Dannce dataset. As shown in the following table, the accuracy drops significantly when the number of viewpoints is less than 4. However, our TR loss achieves consistent improvement in all cases.
>
> |       num of cams      |   6  |   5  |   4  |   3  |   2   |
> |:---------------:|:----:|:----:|:----:|:----:|:-----:|
> | without TR loss | 5.86 | 6.52 | 6.72 | 9.79 | 24.70 |
> | with TR loss    | 3.61 | 4.41 | 4.49 | 6.78 | 18.04 |
>
> > Q4: what is the training time of the current formulation?
>
> R4: The training time for the laboratory mouse pose estimation experiments is around 30-40 minutes. The training time for human pose estimation experiments is around 3-4 hours.
>
> > Q5: What happens if the estimated keypoint from one of the views is erroneous?
>
> R5: If the 2D estimates on one view are erroneous, the TR loss will enforces it to converge to the point that has 3D consistency with other views. Its triangulation confidence will also be lower so that its negative effect is reduced. Therefore, the proposed method achieves relatively robust performance against erroneous 2D estimates.
>
> > Q6: Limitations and social impact have not been discussed.
>
> R6: The limitation of our method is that it requires a 2D detector in giving a relatively accurate initial estimate. This is also a common limitation of 3D pose estimation without 3D supervision. We present some failure cases in the appendix. We will discuss the limitation of our method in the revised paper.
> Regarding the social impact, our method is a novel solution for non-3d supervision pose estimation for both humans and animals, so it is meaningful for both human application and animal behavior analysis. Our method and results have no negative impact on animal conservation and welfare, human health and safety, and social ethics.

---

### Official Review · Reviewer_4Fs8 · 2023-07-06

**Soundness:** 3 good
**Presentation:** 3 good
**Contribution:** 3 good
**Rating:** 7
**Confidence:** 3

**Summary:**

The paper addresses the problem of predicting human or animal joint positions in 3d from a set of posed images (with calibrated cameras). In order to do so the authors propose to train a neural network that leverages the multi-view constraints between images and predicts the 3d positions in and end-to-end fashion.

The contribution of the paper is an objective formulation that doesn't rely on explicit 3d ground truth, but can be supervised from 2d ground truth only and is regularized via a loss on the triangulation residual (in image space, not in 3d). The authors demonstrate that the latter can be achieved by minimizing the smallest singular value of the triangulation matrix; that is, the 3d points never need to be computed explicitly for the loss computation and no 3d grund truth is required for training.

Experimental results demonstrate the effectiveness of the approach for human and mice pose estimation.

**Strengths:**

**S1** 3D ground truth is expensive and hard to obtain compared to 2d annotations. Therefore the proposed framework and losses (requiring 2d ground truth only) address an important and relevant topic for the greated (3d) vision community.

**S2** Experimental results demonstrate that the proposed method performs better than state-of-the-art and also methods that are trained with 3d ground truth. This is a strong contribution and underlines the importance of the unsupervised multi-view constraint in learning.

**S3** The formulation of the the residual minimization is elegant as it does not require to actually triangulate and backproject points for residual computation.

**Weaknesses:**

**W1** Eq. (9) suggests to minimize the smallest singular value of the projection matrix. In order to do so its computation via SVD needs to be differentiable. A discussion on this topic is missing, but is required for the paper to be self consistent. References to related work are also missing, e.g. "Robust Differentiable SVD", Wei Wang, Zheng Dang, Yinlin Hu, Pascal Fua, Mathieu Salzmann, TPAMI 2022.

**W2** A discussion about the relation of the minimized algebraic error (the smallest singular value) and the desired geometric error (the residuals in image space) is missing. Though, it would be helpful for the reader to understand the relation between the minimized error of Eq. (5) and Eq. (9).

**Questions:**

**Q1** Does the number of 3d points / joints to estimate need to be known upfront? Do you aim to predict all joints in each view and how do you handle occlusions?

**Q2** The residual loss in Eq. (9) is formulated for a single 3d point j. How is the loss over all joints formulated? If the optimization is separate per joint, what prevents that two headmap / confidence heads converge onto the same joint?

**Q3** Fig. 3 lists performance numbers for the ablation study and shows that the confidence predictions only have a small influence on the overall prediction performance. Do you have an estimate about the frequency of occlusions per joint in the dataset? I'd imagine that with more occlusions the prediction confidence would become more important in order to be able to triangulate successfully.

*Comment*: In the paragraph starting at line 153 it should read *trivial*, not trial.

**Limitations:**

The paper is missing a discussion about limitations of the approach. I appears that the proposed triangulation loss is of interest not only for the estimation of human / animal joint positions, but also for structure estimation, e.g. jointly reconstructing a point cloud and learning a local interest point detector. However, it remains unclear if the presented approach is applicable to this use-case and what constraints / limitations need to be considered.

Interesting questions to address would be:
a) Is there an upper limit on the number of 3d points that can be handled?
b) How does the method perform if each 3d point is only observed by a (small) subset of images? Can the confidence estimation handle those cases and can the triangulation loss still be formulated over all cameras (Eq. (5-6))? Line 153 ff. mentions that the learnable weights are forced to be > 0 in order to avoid a trivial solution. Is this still applicable in case of only partial observations?

---

> ### Author Rebuttal · Authors · 2023-08-02
>
> We thank the reviewer for the careful and constructive review. To address your questions and concerns:
>
> > Q1:  Eq. (9) suggests to ..., References to related work are also missing, ...
>
> R1: Thanks to the reviewer for recommending "robust differentiable SVD". We will add the reference and discuss this topic in our final paper. Robust differentiable SVD addressed the instability problem of eigendecomposition during deep network training. We will try to use it in the future instead of the current Pytorch-AutoDiff-based SVD.
>
> > Q2: A discussion about the relation of the minimized algebraic error and the desired geometric error is missing...
>
> R2: **The algebraic error Eq. (9) is an approximate geometric error Eq. (5).** The derivation from Eq. (6) to Eq. (9) is the standard procedure of the SVD decomposition solution of the least square optimization problem, which has been strictly proved, so we only need to clarify the relation between Eq. (5) to Eq. (6). By using homogeneous coordinates, for an observed point $x(u,v,t)$ and an estimated point $x’(u’,v’,t’)$ projected from estimated 3D point $X’$ using projection matrix $P_c$ (i.e. $x’=P_c \times X’$), the geometric error penalizes the 2-norm of the geometric error vector and writes $d_{geo}(x,x’) = ||(u/t-u’/t’)^2 + (v/t-v’/t’)||\_2$, where $t’$ is the estimated depth of point $X’$ from the $c$th view, which is usually different from each view. Therefore, directly minimizing $d_{geo}$ usually involves heavy iterative optimization. The algebraic error penalizes the 2-norm of the algebraic error vector and writes $d_{alg}(x,x’) = ||(vt’-tv’)^2 + (u’t-ut’)||\_2$, so $d_{alg} = d_{geo} * t * t’$.  We always set $t=1$. By accumulating the errors from all the points from different views, $d_{alg}$ is not proportional to $d_{geo}$ because $t'$ from different views are inequal. Therefore minizing $d_{alg}$ yields a slightly different results. However, $d_{alg}$ can be solved linearly and is more suitable for end-to-end training.
>
> > Q3: Does the number of 3d points/joints to estimate need to be known upfront? Do you aim to predict all joints in each view and how do you handle occlusions?
>
>
> R3: The number of joints needs to be known for the 2D detector since the number of heatmap head layers should be equal to the joint number. However, TR loss itself does not require the number of joints to be known in advance.
>
> Our final goal is to predict the 3D position of each joint. We first simply filter out the obvious occluded 2D keypoints in each view according to the confidence value output of the 2D detector. For those keypoints with high confidence but still under occlusion, our TR loss can enforces them to converge to a position that is cross-view spatially consistent in 3D.
>
> > Q4:  The residual loss ... How is the loss over all joints formulated? ..., what prevents that two headmap / confidence heads converge onto the same joint?
>
> R4: The TR loss over all joints is formulated as the sum of losses for all joints. The predefined one-to-one relationship between the heatmap channel and its target joint prevents two heatmap heads from converging to the same joint. The same 2D heatmap channel estimates the same joint in different views, while two different 2D heatmap channels are defined to predict different joints in the same view. Therefore, there is no need for additional cross-view matching(association).
>
> > Q5: Fig. 3 ... shows that the confidence predictions only have a small influence ...
>
> R5: We agree with the reviewer that prediction confidence would become more important for cases with more occlusions. Inspired by the comments, we checked the occlusion rate in our data. The average occlusion rate of all joints in the THM mouse dataset is about 27\%, but the occlusion rate of different joints varies a lot (more than 60\% for feet but less than 5\% for ears). The Human3.6M is labeled by 3D ground truth reprojection so that even the occluded joint has a 2D label. We will design a more systematic and detailed ablation study on mouse experiments in the revision.
>
> > Q6: The paper is missing a discussion about limitations of the approach...
>
> R6: The limitation of our method is that it requires a 2D detector in giving a relatively accurate initial estimate. This is also a common limitation of 3D pose estimation without 3D supervision.
> We agree with the reviewer's comment that TR loss may be able to improve some structure estimation methods since it provides better 3D spatial consistency of local points in different views. A potential challenge is that the scene in structure estimation/SFM is much more complex than pose detection, so a better detection model and system-level optimization may be needed to achieve this goal. We will discuss in the future work section.
>
> > Q7: Is there an upper limit on the number of 3d points that can be handled?
>
> Theoretically, there is no upper limit on the number of 3d points that can be handled because our method calculates each point individually.
>
> > Q8: How does the method perform if each 3d point is only observed by a (small) subset of images? ...
>
> R8: We set a threshold of 0.2 on the confidence of the 2D detector output to filter out invisible points (referring to "OpenMMLab Pose Estimation Toolbox and Benchmark"). These points will not be involved in the triangulation process. For other points, we set the learnable triangulation weights to adapt their contributions to triangulation. This weight is not allowed to be 0, because it may lead to a trivial solution, as discussed in L153 of the manuscript.

---

### Author Rebuttal · Authors · 2023-08-10

We thank the reviews for their insightful and valuable comments.
In summary, reviewers are positive about the novelty, formulation, impact, performance, and potential of our method as mentioned:
"address an important topic for the greated (3d) vision community, ... the formulation is elegant. " (**R-4Fs8-S1&S3**),  "achieve substantial improvement on multiple datasets." (**R-ZQc8-S3**), "conduct extensive ablation studies testify the effectiveness of key designs." (**R-3C6c-S1**), "introduce Simple idea with potential." (**R-y7FT-S2**) and "contribute Simple yet effective idea and a new THmouse dataset for laboratory mouse pose estimation." (**R-iymk-S1&S2**).

After carefully analyzing all the reviews, major concerns can be summarized:
1. More rigorous explanation of the mathematical formulation. (**R4Fs8-W1&W2 and Ry7FT-W3**)
2. More in-depth analysis of the robustness. (**R-4Fs8-Q3, R-ZQc8-Q2, R-y7FT-W2&Q1 and R-iymk-Q4**)
3. More thorough comparisons and demonstrations. (**R-3C6c-W2, R-y7FT-W1, R-iymk-W1&W2 and R-ZQc8-W2**)
4. Missing detailed discussion on the limitation. (**R-4Fs8, R-ZQc8, R-3C6c and R-iymk**)

We simply summarize the response to each major concern at here.
For more detailed illustrations and experimental results, please refer to the reviewer-specific response letters.
1. For the description of the mathematical formulation, the algebraic error is indeed an approximation of the geometric error in the standard SVD algorithm. We will clarify the description and add a more detailed explanation in the revision. Moreover, although we use Pytorch auto diff in our implementation of the TR Loss, we regard the mentioned "robust differentiable SVD" method as an interesting direction to further improve the performance of the implementation, and we will discuss this method in both the related work and the future work section.
2. For a more in-depth analysis of the robustness, we analyze in detail the robustness of our method under different situations like occlusions, outliers, and partial observations. Please refer to the detailed response to the corresponding questions. Note that the overall improvement of our method mainly comes from the key design: optimizing the 2D keypoint detector with a more "global 3D aware" TR loss, which enhances multi-view 3D consistency in an unsupervised manner.  We also ablate the performance under different view number setups (from 2 to 6), which also demonstrate the effectiveness of our method when compared with other SOTA methods.
3. For more thorough comparisons and demonstrations, we provide an additional video result (will provide more in the revision), ablate the TR Loss with different 2D detectors and also compare with other SOTA mouse 3D pose estimation methods. Due to the time limit, we are not able to add experiments on additional datasets, and with different objects, we will add more experiments in the revision.
4. Missing detailed discussion on the limitation: we have discussed in detail the limitation of our method in the reviewer-specific response letter, and we will clarify this in the revision according to the review, the response, and the failure cases we provide in the appendix.

---

### Decision · Program_Chairs · 2023-09-21

**Decision:**

Accept (poster)

**Comment:**

After discussion between the authors and the reviewers, all reviewers are inclined to accept the paper. The AC agrees. Given the good reviews and discussion, the AC strongly encourages the authors to incorporate all changes from the discussion into the final version of the paper.